# 3D Multiple Triangular Prisms for Highly Sensitive Non-Contact Mode Triboelectric Bending Sensors

**DOI:** 10.3390/nano12091499

**Published:** 2022-04-28

**Authors:** Gi Hyeon Han, Sun Woo Kim, Jin Kyeom Kim, Seung Hyun Lee, Myeong Hoon Jeong, Hyun Cheol Song, Kyoung Jin Choi, Jeong Min Baik

**Affiliations:** 1School of Advanced Materials Science and Engineering, Sungkyunkwan University (SKKU), Suwon 16419, Korea; ghhan@unist.ac.kr (G.H.H.); kimsunwoo0917@gmail.com (S.W.K.); jingyeom0825@naver.com (J.K.K.); samysh@g.skku.edu (S.H.L.); hcsong@kist.re.kr (H.C.S.); 2School of Materials Science and Engineering, Ulsan National Institute of Science and Technology (UNIST), Ulsan 44919, Korea; mhjeong0102@unist.ac.kr (M.H.J.); choi@unist.ac.kr (K.J.C.); 3KIST-SKKU Carbon-Neutral Research Center, Sungkyunkwan University (SKKU), Suwon 16419, Korea; 4Center for Electronic Materials, Korea Institute of Science and Technology (KIST), Seoul 02792, Korea

**Keywords:** bidirectional bending sensor, self-powered sensor, triboelectric, non-contact mode, less sensitive to strain

## Abstract

Here, a highly sensitive triboelectric bending sensor in non-contact mode operation, less sensitive to strain, is demonstrated by designing multiple triangular prisms at both sides of the polydimethylsiloxane film. The sensor can detect bending in a strained condition (up to 20%) as well as bending direction with quite high linear sensitivity (~0.12/degree) up to 120°, due to the electrostatic induction effect between Al and poly (glycerol sebacate) methacrylate. Further increase of the bending angle to 135° significantly increases the sensitivity to 0.16/degree, due to the contact electrification between them. The sensors are attached on the top and bottom side of the proximal interphalangeal and wrist, demonstrating a directional bending sensor with an enhanced sensitivity.

## 1. Introduction

Recently, flexible sensor systems, inspired by human skin’s ability to sense a variety of stimuli, have been widely explored, with wide applications such as interactive wearable devices, smart electronic devices, intelligent robots, and health monitoring systems [1,2,3,4]. Up to now, according to the sensing mechanism, various sensing technologies based on the capacitor type, resistor type, and piezoelectric type have been developed and successfully proved the ability to sense various and complex stimuli such as pressure, strain, motion, etc. [5,6,7,8]. Among them, capacitive sensors have been intensively studied because they have many advantages such as long-term stability, low power consumption, fast response, and low recovery time [9,10,11]. However, their further applications are still restricted by the relatively low sensitivity and the sensing range, as well as the requirement of an external power source.

Recently, triboelectric nanogenerators (TENGs), in which the electric outputs were generated based on the coupling effect of contact electrification and electrostatic induction, have attracted much attention for use as power sources for small electronic devices [12,13,14,15]. They have many advantages such as high output, low cost, and mechanical robustness, which successfully offered themselves to several self-powered systems. Apart from the power supply, triboelectric-type sensors, implemented with high sensitivity to external stimuli, low cost fabrication, and high applicability of design, have been also intensively studied for sensing momentary stimuli in flexible touch sensors, electronic skins, biomedical devices, and smart security systems [16,17,18,19]. Furthermore, they can normally operate without external power supply, which has enormous potential in Internet of Things (IoT) and biomedication applications [20,21,22,23,24,25]. However, most sensors are based on the contact-separation mode occurring between two contacted materials, limiting the sensing range and accurate and stable sensing in various application circumstances.

Here, a triboelectric-type sensor based on the non-contact mode is demonstrated to show high sensitivity to bending motion up to 120° while being less sensitive to strain, and provides the information of bending direction, by designing multiple triangular prisms at both sides of the polydimethylsiloxane (PDMS) film. Under normal bending force, the sensor generates up to 1.2 V at low angles (<120°), corresponding to a sensitivity of about 0.12/degree, which drastically increases to 1.8 V (~0.16/degree) at the high angle of 135°. The frequency- and bending-angle-dependent output voltage measurement reveals that at low angles, the output signals are generated via only electrostatic induction effect between Al and poly (glycerol sebacate) methacrylate (PGSm), while at high angles, the output is drastically increases due to the coupling effect of the contact electrification and electrostatic induction effect. The sensor performance in strained condition shows that it is less sensitive to the strain because the magnitude of the output voltage is determined by the gap distance, which linearly increases from the bottom to the top region of triangles. Finally, the sensor is attached on the top and bottom side of the proximal interphalangeal and wrist, demonstrating a directional bending sensor with an enhanced sensitivity.

## 2. Materials and Methods

### 2.1. Fabrication of 3D-Printed Bending Sensors

The bending sensor was fabricated using 3D SLA (stereolithography) method using PGSm (Poly (glycerol sebacate) methacrylate)-based resin with good elastic properties. The SLA 3D printer is shown in Appendix A. The 3D printing parameters are shown in Appendix A. The printed sensor was soaked in IPA with sonication for 1 day to sufficiently rinse the uncured resins. The resultant sensor is shown in Figure 1a. A master mold was fabricated to make the triangular prism structure, leaving the PGSm material on a side of PDMS, and 500 nm-thick Al was deposited onto the PGSm at an oblique angle by e-beam evaporation. Another triangular prism structure with Al was also fabricated at the opposite side of the PDMS, attached on the PDMS. Al-tape was attached so that the electrodes on both ends of the sensor were connected. The bidirectional bending sensor has a depth of 1 cm and a length of about 1.1 mm. The overall thickness was only 3 mm.

### 2.2. Sensor Evaluation

A Tektronix DPO 3052 Digital Phosphor oscilloscope and a low-noise current preamplifier (model no. SR570, Stanford Research Systems, Inc., Sunnyvale, CA, USA) were used for the measurement of the output voltages and current densities of the TENGs. A pushing tester (Labworks Inc., Costa Mesa, CA, USA, model no. ET-126-4) was used to apply a vertical force to the TENG. A bending machine (Bending tester ZB-100) was used to apply a concave and convex bending to the sensors. The output voltage signal was measured using an oscilloscope. The electric output signal measurement instrument setup by the bending motion is shown in Appendix A. The capacitance due to bending of the sensor was measured using a precision LCR meter (Hioki 5355, Hioki, Nagano, Japan) and bending machine under ambient conditions. The directions of flat shape, concave bending, and convex bending are shown in Appendix A.

## 3. Results

The schematic diagram for the fabrication of the triboelectric-type bending sensor is shown in Figure 1a; detailed information is described in Methods. The sensor consists of multiple triangular prisms at both sides of the polydimethylsiloxane (PDMS) film supporter. To investigate the capability of the sensor as a practical bending sensor, the capacitance change was measured with bending angles up to 135°, plotted in Figure 1b. As the bending angle of the sensor was increased from 0° to 135°, the capacitance was increased from 0.6 pF to 3.8 pF. This may be ascribed to the decrease of the air gaps between the triangles, as shown in the inset of the Figure 1b [26]. Conversely, as the bending angle was decreased, the capacitance decreased and recovered its original value, indicating that plastic deformation did not occur in constituent materials of the sensor. The bending test was done for 2000 s with a frequency of 1 Hz; it clearly showed that there was no significant change in the capacitance value, showing good stability of the sensor, as shown in Figure 1c.

Prior to evaluate the sensing performance, a TENG consisting of poly (glycerol sebacate) methacrylate (PGSm)/Al (top layer) and Al (bottom layer), as in the inset of Figure 1d, was fabricated and electric outputs were measured under a cycled compressive force of around 30 N at a frequency of 3 Hz. The TENG has an active area of 1 cm × 1 cm and gap distance of 1 mm, and four springs with lengths of 5 mm in each corner. The framework of the TENG was fabricated using a fused deposition modeling (FDM)-based 3D printer with acrylonitrile butadiene styrene (ABS) filament. When the top layer was pressed onto the bottom layer and released, an instantaneous positive potential (V_oc_) of about 15 V at the open circuit condition was generated, followed by a negative signal (28 V) when the top layer was pressed again. This implies that the charged species (which may include electrons) are transferred between two materials intensively, as indicated by strong electrical signals, and the Al acts as a positively charged layer. The TENG also generated an instantaneous output current (I_sc_) of about 21 μA at a short circuit condition. The distance between the Al and the PGSm when the external force was applied on the top layer was controlled to be from 0.2 mm to 2 mm, which are the shortest distances between the two layers during the measurement. At a distance of 0.2 mm apart, the V_oc_ was drastically decreased to about 5 V and it continuously, almost linearly decreased with the distance, as shown in Figure 1f. Finally, about 1 V was measured at 2.0 mm. The decrease in the output voltage may be ascribed to the decrease of the electrostatic force between the two materials as the separation distance increases. However, it is quite high enough as a sensor signal in non-contact mode.

To maximize the sensing performances, various parameters such as number of the triangles, angle between triangles, and height of the triangles, were optimized by measuring electric outputs of the sensor having multiple triangular prisms at one side of the PDMS, as shown in Appendix A. Appendix A shows the output voltages of the sensor with the number (2, 3, 4, and 5) of the triangular prisms. As the number of triangular prisms increased, the voltages were increased and about 1.0 V was generated at 4 or 5 prisms. The increase of the generated potentials may be explained via the alignment of a pair of opposite charges in same direction, as shown in Appendix A. Figure 2a shows the output voltage of the sensor with different angles between triangles. The height of the triangles was fixed to be about 1.4 mm and the bending angle was about 135°. At 45°, an instantaneous voltage of about 0.3 V was generated and as the angle was increased to 90°, voltage also increased to about 0.9 V. The increase of the voltage with the angle between triangles implies that the generated potential was strongly dependent on the gas distance between the triangles. Actually, the open-circuit voltage is linearly proportional to the gap distance because the total capacitance in the device is inversely proportional to the gap distance [27]. At a fixed angle of 90°, the height of the triangles was changed and the output voltages were measured under same condition. As the height was increased, the output voltages increased, ascribed to the increase of the contacted surface area. The sensor was then operated continuously for 5000 s at a frequency of 1 Hz. As shown in Appendix A, it generated an instantaneous output voltage of about 1.45 V, where there was no change even after 5000 cycles, demonstrating its excellent durability.

A bending sensor having multiple triangular prisms at both sides of the PDMS was fabricated and the instantaneous electric outputs were measured at a bending angle of 135°, compared with the bending direction, as shown in Figure 2c,d. When the sensor was under concave bending, an instantaneous output voltage of about 0.8 V was generated. The sensitivity S can be defined to be S = (V_b_ − V_o_)/V_o_, where V_b_ and V_o_ denote the output voltage with and without the bending motion, respectively. Here, in our instrument, V_o_ is usually measured as approximately 50 mV. This indicates that the sensitivity of the sensor is about S = 0.12/degree, showing quite high sensitivity, compared with others reported so far [28,29,30,31,32]. The expanded view of the signal showed that it consisted of two pulses, i.e., a positive peak, followed by a negative peak. However, under convex bending, the negative peak was followed by the positive peak, in which case the magnitude of output voltage was almost the same. This can be explained via the direction of dipole moment due to the positive and negative charges generated by the contact electrification between Al and PGSm. Under concave bending, the triangles at the top side of PDMS are contacted and separated by the bending motion. By the contact between Al and PGSm, negative charges are transferred from Al to PGSm, forming alternating negative and positive charges in turn. This generates an electrical potential which can induce electron flow between two Al electrodes when the triangles are separated. However, under convex bending, the triangles at the other side are contacted and separated by the bending motion. At the bending motion, charges distribution is reversed, resulted in opposite direction of the electron flow. These results clearly show that the sensor can provide exact bending information, as well as high sensitivity.

For further investigation of the effect of the bending condition on the sensing performance, the output voltages were measured as a function of frequency and bending angle, plotted in Figure 3a,b, respectively. The photos of the sensors with bending angle are shown in Appendix A. Under 0.5 Hz, an output voltage of about 0.82 V was measured and as the frequency was increased, voltage increased to about 1.75 V at 2.5 Hz. In general, the enhancement in the electric output by the increasing frequency may be explained by the higher moving speed of the two materials [33]. The transferred charge density increases with contact force, which also increases with the speed because of the increase of the impulse (*I* = *F*t = 2 m*v*). However, we fixed the bending speed to be about 270°/s. Another reason may be large loss of charges created on the surface of both contacted materials. The frequency-dependent charge density measurement showed that the charge density was significantly decreased with mean free times between the contacts at low frequencies of less than 10 Hz [34]. This implies that the rate of the charge loss caused by the charge dissipation is high.

Figure 3b shows the output voltage of the sensor with a bending angle from 30° to 135°. Bending angle was analyzed by the photos in the inset of the Figure 3b. At the bending angle of 30°, the instantaneous voltage of 0.5 V was measured and as the angle was increased, voltage increased to about 1.24 V at the bending angle of 120°. The sensitivity of the sensor with respect to the bending angle was about 0.12/degree, as shown in Figure 3c. The voltage almost linearly increased with the bending angle. Further increase in the bending angle to 135° yielded drastic increase to about 1.74 V, corresponding to the sensitivity of about 0.16/degree. This shows that the output generation mechanism at high angles (>120°) is quite different with that at low angles (<120°). The detailed investigation of the photos in Appendix A shows that the triangles were not contacted at low angles and fully contacted at 135°. This indicates that the output generation mechanism at low angles may not be due to the contact electrification. In general, the output power generation mechanism in TENG is due to the coupling effect of contact electrification and electrostatic induction [35,36]. However, at low angles, there was no physical contact between the triangles, indicating that the power generation was due to the electrostatic induction. At 135°, the voltage was significantly increased via the contact electrification. The sensor performance was also evaluated under the condition of tensile strains up to 20%, plotted in Figure 3d. Here, the bending angle was fixed to be about 120°. With no strain applied, it generated an instantaneous output voltage of about 1.4 V, which is similar with the data in Figure 3b. As the strain was increased, it seems that the output voltage gradually increased to about 1.7 V at 20%; however, the increase was not significant. Previously, numerical calculations proved that the voltage was maximized at the optimized distance of less than 1 mm between two contacted materials, which shifted to a larger value with the speed of the moving plate [37,38]. Here, the distance between the tip of the triangles was optimized to be about 2.8 mm, indicating that the generated voltage was determined by the gap distance varying from 0 (bottom region) to 2.8 (top region) mm. At 20% strain, the average distance between the triangles was increased by 0.28 mm. This shows that the change of the output voltages in strained condition was not significant, evident by the output voltages shown in Figure 1f. It should be noted that other small peaks between the large peaks were observed, unlike the data in Figure 2a,b. This may be due to the difference of the triangles on both surfaces. Appendix A shows the output voltages measured when the sensor was strained up to 20% and released. It clearly shows that the output voltages of about 0.1 V were measured at 20% strain.

Finally, the sensor having triangular prisms at one side was attached on the top and bottom side of the proximal interphalangeal (PIP) joint of a human finger, as shown in Figure 4a,b. When one bends the finger, the sensor attached on the top side generated 0.05 to 0.3 V, up to a bending angle of 105°. As expected, the voltage almost linearly increased with the bending angle. When the sensor was attached on the top side, the output voltage significantly increased to the range of 0.5 to 1.25 V under the same motion. Attaching the sensor to the bottom of the PIP and applying bending reduces the spacing of the triangles and induces electrostatic induction to generate a signal. On the other hand, if a sensor is attached to the upper part and bending is applied, the gap distance increases, and a signal is generated by electrostatic induction and a relatively small signal is generated. Due to electric field strength being inversely proportional to distance, the magnitude of the electric field change that occurs as the gap distance decreases is larger than the electric field change that occurs as the gap distance increases.

To demonstrate the capability of the sensor to identify a particular motion, the sensor was firmly attached on the top and bottom side of wrist. The output voltages were measured by repeatedly bending the wrist upward and downward at a frequency of about 1 Hz, as shown in Figure 4c. When the sensor was attached on the top side, the motion generated a voltage signal of around 1.0 V, followed by another signal of about 0.2 V. The former voltage consisted of two pulses, i.e., a positive peak, followed by a negative peak, as shown in the inset of Figure 4c. When the sensor was attached on the bottom side, the output performance was quite similar, except that the former voltage consisted of a negative peak, followed by a positive peak. Thus, according to the magnitude of the signal and the order of positive/negative peak, the sensor can provide information on bending direction. These advantages offer great potential for electronic and mechanical platforms such as self-powered smart skins and bending sensors that enable bidirectional measurements.

## 4. Conclusions

In summary, a highly sensitive, strain-insensitive bending sensor based on the noncontact operating mode was demonstrated. The sensor was designed to have multiple triangular prisms at both sides of the polydimethylsiloxane film, based on the SLA method. It was quite stable during repeated bending tests, up to 2000 times. In order to maximize the sensing performance, the height of triangles and the angle between triangles were optimized. The frequency- and bending-angle-dependent output voltage measurement showed that at low angles (<120°), the output voltage, generated via only electrostatic induction effect between Al and PGSm, almost linearly increased with the bending angle. The output voltages were drastically increased via the contact electrification at a high angle (135°). The sensor performance in strained condition also showed that it was less sensitive to strain because the magnitude of the output voltage was determined by the gap distance, which linearly increased from the bottom to the top region of triangles. Finally, the sensor with triangles at one side of PDMS was attached on the top and bottom side of the proximal interphalangeal and wrist, demonstrating a directional bending sensor with an enhanced sensitivity.

## Figures and Tables

**Figure 1 nanomaterials-12-01499-f001:**
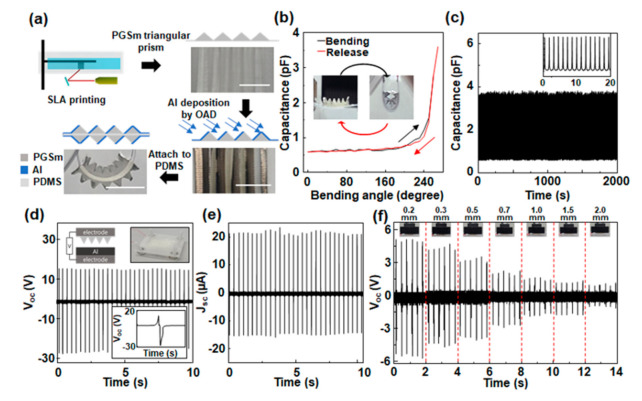
(**a**) The schematic diagram for the fabrication of triboelectric-type bending sensor. Scale bar: 1 cm. (**b**) Capacitance change of triboelectric-type bending sensor and (**c**) stability test. (**d**) Out-put open-circuit voltage and (**e**) short-circuit current of PGSm/Al TENG. (**f**) Output voltages of PGSm/Al TENG with various gap distance from 0.2 mm to 2.0 mm.

**Figure 2 nanomaterials-12-01499-f002:**
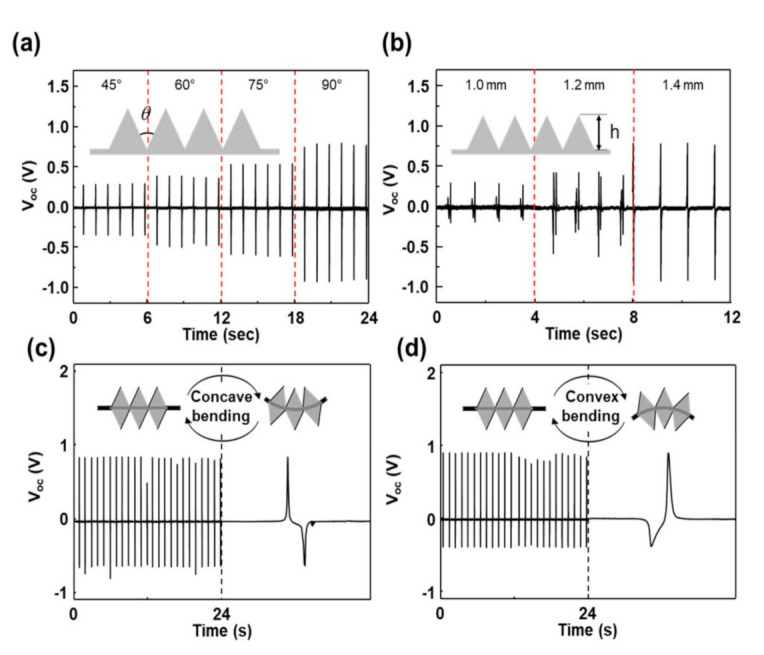
(**a**) Output voltages of the sensor with different angles between triangles. (**b**) Output voltages of the sensor with different height. Output voltages of both sides sensors with (**c**) concave bending and (**d**) convex bending.

**Figure 3 nanomaterials-12-01499-f003:**
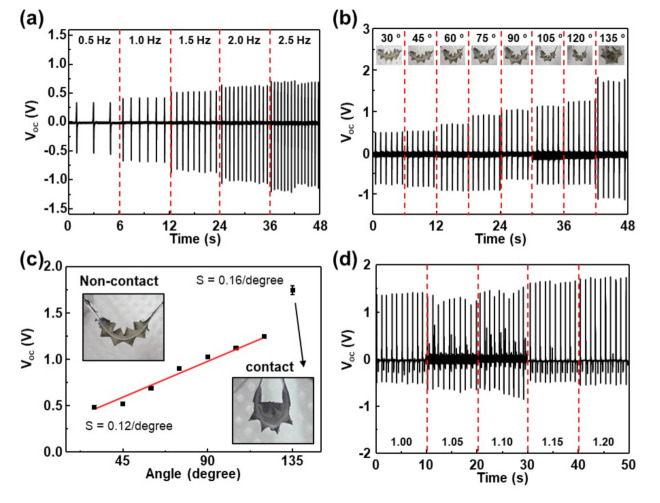
(**a**) Output voltages of the sensor with different frequency. (**b**,**c**) Output voltages of the sensor with bending angles and sensitivity. (**d**) Output voltages of sensor with bending under strain from 1.00 to 1.20.

**Figure 4 nanomaterials-12-01499-f004:**
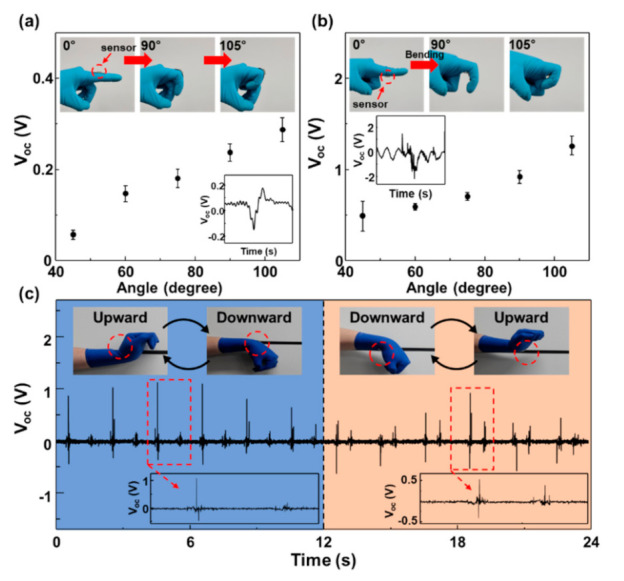
Output voltages of the attached on the (**a**) top and (**b**) bottom side of the proximal interphalangeal (PIP) with bending. (**c**) Output voltages of the attached on the top and bottom side of the wrist and detecting bending directions.

## Data Availability

The data are available on request from the corresponding author.

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
