# Peer review of "3D Multiple Triangular Prisms for Highly Sensitive Non-Contact Mode Triboelectric Bending Sensors"

_nanomaterials, 2022, doi:10.3390/nano12091499_

Round 1

Reviewer 1 Report

In this work, the authors reported a triboelectric-type sensor based on the non-contact mode by designing multiple triangular prisms at both sides of the polydimethylsiloxane (PDMS) film. It shows high sensitivity to bending motion up to 120 ° but less sensitive to strain. Experiments show that the sensor’s sensitivity difference at different angles is due to the electrostatic induction effect and contact electrification effect of Al and PGSm. The sensor is attached on the top and bottom side of the proximal interphalangeal and wrist, demonstrating a directional bending sensor with an enhanced sensitivity. The manuscript is logical and orderly. This manuscript is recommended for publication in Nanomaterials after Minor Revision by properly addressing the following issues:

  1. Please clearly indicate the meaning of the illustrations shown in Figure 1a, and mark the respective scale bars in the figures.
  2. Please explain why the open-circuit voltage of TENG in Figure 1f decreases as the distance between Al and PGSm increases.
  3. Do different numbers of triangular prisms affect the output performance of TENG?
  4. Some references about non-contact sensors and triboelectric sensors are suggested to be also cited: Adv. Funct. Mater.2021, 31, 2105110; Adv. Funct. Mater. 2021, 31, 2100940; ACS Nano, 2022, 10.1021/acsnano.1c11096.
  1. What is the maximum strain of this angle sensor?

Author Response

Reviewer’s comments:

Reviewer#1: In this work, the authors reported a triboelectric-type sensor based on the non-contact mode by designing multiple triangular prisms at both sides of the polydimethylsiloxane (PDMS) film. It shows high sensitivity to bending motion up to 120 ° but less sensitive to strain. Experiments show that the sensor’s sensitivity difference at different angles is due to the electrostatic induction effect and contact electrification effect of Al and PGSm. The sensor is attached on the top and bottom side of the proximal interphalangeal and wrist, demonstrating a directional bending sensor with an enhanced sensitivity. The manuscript is logical and orderly. This manuscript is recommended for publication in Nanomaterials after Minor Revision by properly addressing the following issues:

Point 1: Please clearly indicate the meaning of the illustrations shown in Figure 1a and mark the respective scale bars in the figures.

Response 1: We strongly appreciate reviewer’s valuable and helpful comments. We added some explanations every step to Figure 1a. and scale bars to the three photos.

Figure 1. The schematic diagram for the fabrication of triboelectric-type bending sensor. Scale bar: 1 cm.

Point 2: Please explain why the open-circuit voltage of TENG in Figure 1f decreases as the distance between Al and PGSm increases.

Response 2: As the reviewer may know, the distance between Al and PGSm means the shortest distances between the two layers during the measurement. In electrostatics, the electrical force between two charged objects is inversely related to the distance of separation between the two objects. Thus, increasing the separation distance between objects decreases the force of attraction between the objects. This make the charge density on the surface of the Al decrease, resulted in a decrease of the generated voltage.

We added the following sentence to clarify the influence of the distance on the generated voltage to page 4.

“The decrease in the output voltage may be ascribed to the decrease of the electrostatic force between the two materials as the separation distance increases.”

Point 3: Do different numbers of triangular prisms affect the output performance of TENG?

Response 3: We fabricated the sensor with the number (2, 3, 4, and 5) of the triangular prisms and the open circuit voltages were measured, as plotted in Figure 2. As the number of triangular prisms increased, the voltages were increased and about 1.0 V was generated at 4 or 5 prisms.

Figure 2. Open circuit voltages of the sensor with the number (2, 3, 4, and 5) of the triangular prisms

Point 4: Some references about non-contact sensors and triboelectric sensors are suggested to be also cited: Adv. Funct. Mater.2021, 31, 2105110; Adv. Funct. Mater. 2021, 31, 2100940; ACS Nano, 2022, 10.1021/acsnano.1c11096.

Response 4: According to the reviewer’s comment, we added the following references into page 1 (line 45).

[23] Sohel Rana, S.M.; Zahed, M.A.; Rahman, M. T.; Salauddin, M.; Lee, S.H.; Park, C.; Maharjan, P.; Bhatta, T.; Shrestha, K.; Park, J.Y.; Cobalt-Nanoporous Carbon Functionalized Nanocomposite-Based Triboelectric Nanogenerator for Contactless and Sustainable Self-Powered Sensor Systems. Adv. Funct. Mater.2021, 31, 2105110.

[24] Xiang, S.; Liu, D.; Jiang, C.; Zhou, W.; Ling, D.; Zheng, W.; Sun, X.; Li, X.; Mao, Y.; Shan, C. Liquid-Metal-Based Dynamic Thermoregulating and Self-Powered Electronic Skin. Adv. Funct. Mater.2021, 31, 2100940.

[25] Li, X.; Zhu, P.; Zhang, S.; Wang, X.; Luo, X.; Leng, Z.; Zhou, H; Pan, Z.; Mao, Y. A Self-Supporting, Conductor-Exposing, Stretchable, Ultrathin, and Recyclable Kirigami-Structured Liquid Metal Paper for Multifunctional E‑Skin. ACS Nano, 2022, 10.102/acsnano.1c11096.

Point 5: What is the maximum strain of this angle sensor?

Response 5: In the manuscript, the sensor was strained up to 20 %. Further increase in the strain to 25 % broke the sensor, as shown in Figure 3. The figure shows that the film was torn at the area between the triangular prisms. The maximum strain of poly(glycerol sebacate) methacrylate (PGSm) and PDMS, which are the materials constituting the sensor, was reported to be larger than this. As a characteristic of the sensor structure, the pressure concentration is generated by the method, and the maximum strain is lower than that of the reported literature.

Figure 3. Pictures of the sensors with strained condition up to 25 %

Reviewer 2 Report

This is good submission presenting the bending sensor operating from triboelectrification between Al and poly (glycerol sebacate) methacrylate (PGSm). The pyramids were deposted on polydimethylsiloxane (PDMS). One plane of pyramid was covered with Al to ensure the triboelectrification upon bending. Althoug the manuscript is well written, some clarifications and improvments are required.
1) Authors should include examples of friction driven TENG devices which already are presented in literature and does not need separable parts, for example: Adv. Funct. Mater. 2015, 25, 3688 and Adv. Mater. Technol. 2021, 2100163.

2) Authors are claiming on pg. 3 (line 116-117) and pg. 5. (line 159-162) that the electrons are transferred from Al to PGSm. However, it is widely proven that the electron transfer is not the mechanism for polymer triboelectrification. I kindly guide authors to study some works related to in-depth study of polymer triboelectrification: Science, 2011, 333, 308–312; Angew. Chem. 2012, 124, 4927 –4931; Mater. Horiz., 2020,7, 520-523;  J. Am. Chem. Soc. 2012, 134, 7223−7226; Macromol. Mater. Eng. 2020, 305, 1900638; Nature Reviews Chemistry volume 3, pages 465–476 (2019); Energy Environ. Sci., 2019,12, 2417-2421; ACS Appl. Mater. Interfaces 2021, 13, 37, 44935–44947; J. Phys. Chem. C 2018, 122, 16154−16160.

3) It is not clear from the manuscript how does the number of triangles influence the output. Are surface charges from middle pyramids are actually inducing charges on working electrodes?

4) It is not clear how parasitic signal from frictional triboelectric charge  forming between rubber glove and sensor device was excluded from measurement presented in Fig. 4? Seems that the measured signal from firction between device and rubber is mistaken with device output as it was described here recently: Adv. Mater. 2020, 2002979.

Author Response

April 18, 2022

Katarina Nesovic
Assistant Editor
MDPI Belgrade

Ms. Ref. No.: nanomaterials-1657907
Title: 3D multiple triangular prisms for highly-sensitive non-contact mode triboelectric bending sensors Nanomaterials

Dear Katarina Nesovic,

We believe that the manuscript qualifies as a paper in Nanomaterials. We think that we fully responded to the concerns and comments of the reviewers. Our reasons are as follows:

Reviewer’s comments:

Reviewer#2: This is good submission presenting the bending sensor operating from triboelectrification between Al and poly (glycerol sebacate) methacrylate (PGSm). The pyramids were deposted on polydimethylsiloxane (PDMS). One plane of pyramid was covered with Al to ensure the triboelectrification upon bending. Althoug the manuscript is well written, some clarifications and improvments are required.

Point 1: Authors should include examples of friction driven TENG devices which already are presented in literature and does not need separable parts, for example: Adv. Funct. Mater. 2015, 25, 3688 and Adv. Mater. Technol. 2021, 2100163.

Response 1: We strongly appreciate reviewer’s valuable and helpful comments. According to the reviewer’s comment, we added the following references into page 1 (line 45).

[15] 15. Sutka, A.; Malnieks, K.; Linarts, A.; Lapcinskis, L.; Verners, O.; Timusk, M. Triboelectric Laminates with Volumetric Electromechanical Response for Mechanical Energy Harvesting. Adv. Mater. Technol. 2021, 6, 2100163.

[19] Yi, F.; Lin, L.; Niu, S.; Yang, P.K.; Wang, Z.; Chen, J.; Zhou, Y.; Zi, Y.; Wang, J.; Liao, Q.; Zhang, Y.; Wang, Z.L. Stretcha-ble-Rubber-Based Triboelectric Nanogenerator and Its Application as Self-Powered Body Motion Sensors. Adv. Funct. Mater. 2015, 25, 3688–3696

Point 2: Authors are claiming on pg. 3 (line 116-117) and pg. 5. (line 159-162) that the electrons are transferred from Al to PGSm. However, it is widely proven that the electron transfer is not the mechanism for polymer triboelectrification. I kindly guide authors to study some works related to in-depth study of polymer triboelectrification: Science, 2011, 333, 308–312; Angew. Chem. 2012, 124, 4927 –4931; Mater. Horiz., 2020,7, 520-523; J. Am. Chem. Soc. 2012, 134, 7223−7226; Macromol. Mater. Eng. 2020, 305, 1900638; Nature Reviews Chemistry volume 3, pages 465–476 (2019); Energy Environ. Sci., 2019,12, 2417-2421; ACS Appl. Mater. Interfaces 2021, 13, 37, 44935–44947; J. Phys. Chem. C 2018, 122, 16154−16160.

Response 2: Although the principle of contact-electrification has long been studied, but whose mechanisms are still unclear. Recently, surface-states and electron cloud/potential models have been employed to explain the contact electrification mechanism.[1-3] However, these models are limited to metal-semiconductors and metal-insulators, i.e.: they are not currently compatible with metal-polymers, as the reviewer commented.[4-5] In general, for polymers having “mobile” ions, ion transfer mechanism has been proposed to explain the triboelectrification, confirmed by surface analysis tools such as x-ray photoelectron spectroscopy.[6] The material transfer mechanism due to mechanical friction was also observed in contact electrification involving polymers[7-15]. In this manuscript, the contacted materials are the Al and PGSm. The PGS-based polymers, synthesized via pre-polycondensation and then cross-linked to form PGS elastomers with covalent three-dimensional networks, are mechanically very stable in view of the polymer's physical properties and are not easily damaged by friction or impact[16-18]. This may reduce the possibility of the ion and material transfer mechanism for the triboelectrification. But some works related to in-depth study of polymer triboelectrification are needed, as the reviewer commented. Lots of investigation for the triboelectrification from materials’ aspect may strongly suggest that more than one mechanism is involved in the charge transfer.

 In this manuscript, we added the following sentence to clarify the charge transfer phenomena between the metal and the insulator to page 3.

“Although the contact-electrification mechanism is not still clear, according to the electron transfer mechanism occurring during metal-insulator contacts, the Al releases electrons to PGSm.”

Reference

[1] Lowell, J. Contact electrification of metals. J. Phys. D: Appl. Phys., 1975, 8, 53-63.

[2] Harper, W. R. The Volta effect as a cause of static electrification. Proc. R. Soc. A, 1951, 205, 83-103.

[3] Duke, C.B.; Fabish, T. J. Contact electrification of polymers: A quantitative model. J. Appl. Phys., 1978, 49, 315-321.

[4] Wu, J.; Wang, X.; Li, H.; Wang, F.; Yang W.; Hu, Y. Insights into the mechanism of metal-polymer contact electrification for triboelectric nanogenerator via first-principles investigations. Nano Energy, 2018, 48, 607-616.

[5] Xu, C.; Zi, Y.; Wang, A.C.; Zou, H.; Dai Y.; He, X.; Wang, P.; Wang Y.-C.; Feng, P.; Li, D.; Wang, Z.L. On the Electron-Transfer Mechanism in the Contact-Electrification Effect. Adv. Mater., 2018, 30, 1706790.

[6] Law, K.-Y.; Tarnawskyj, I.W.; Salamida, D.; Debies T. Investigation of the Contact Charging Mechanism between an Organic Salt Doped Polymer Surface and Polymer-Coated Metal Beads. Chem. Mater. 7, No. 11, 2090-2095 (1995).

[7] Baytekin, H.T.; Patashinski, A.Z.; Branicki, M.; Baytekin, B.; Soh, S.; Grzybowski, B.A. The mosaic of surface charge in contact electrification. Science 2011, 333, 308-312.

[8] Baytekin, H.T.; Baytekin, B.; Incorvati, J.T.; Grzybowski, B.A. Material transfer and polarity reversal in contact charging. Angew. Chem. Int. Ed. 2012, 51, 4843-4847.

[9] Šutka, A.; Linarts, A.; Mālnieks, K.; Stiprais, K.; Lapčinskis, L. Dramatic increase in polymer triboelectrification by transition from a glassy to rubbery state. Mater. Horiz. 2020, 7, 520-523.

[10] Baytekin, B.; Baytekin, H.T.; Grzybowski, B. A. What really drives chemical reactions on contact charged surfaces? J. Am. Chem. Soc. 2012, 134, 7223-7226.

[11] Lapčinskis,L.; Mālnieks,K.; Blūms,J.; Knite,M.; Oras,S.; Käämbre,T.; Vlassov, S.; Antsov, M.; Timusk, M.; Šutka, A. The Adhesion‐Enhanced Contact Electrification and Efficiency of Triboelectric Nanogenerators. Macromol. Mater. Eng. 2019, 305, 1900638.

[12] Lacks, D.J.; Shinbrot, T. Long-standing and unresolved issues in triboelectric charging. Nat. Rev. Chem. 2019, 3, 465-476.

[13] Šutka, A.; Mālnieks, K.; Lapčinskis, L.; Kaufelde, P.; Linarts, A.; Bērziņa, A.; Zābels, R.; Jurķāns, V.; Gorņevs, I.; Blūms, J.; et al. The role of intermolecular forces in contact electrification on polymer surfaces and triboelectric nanogenerators. Energy Environ. Sci. 2019, 12, 2417-2421.

[14] Sherrell, P.C.; Sutka, A.; Shepelin, N.A.; Lapcinskis, L.; Verners, O.; Germane, L.; Timusk, M.; Fenati, R.A.; Malnieks, K.; Ellis, A.V. Probing Contact Electrification: A Cohesively Sticky Problem. ACS Appl. Mater. Interfaces 2021, 13, 44935-44947.

[15] Pandey, R.K.; Kakehashi, H.; Nakanishi, H.; Soh, S. Correlating material transfer and charge transfer in contact electrification. J. Phys. Chem. C 2018, 122, 16154-16160.

[16] Pashneh-Tala, S.; Owen, R.; Bahmaee, H.; Rekstyte, S.; Malinauskas, M.; Claeyssens, F. Synthesis, Characterization and 3D Micro-Structuring via 2-Photon Polymerization of Poly(glycerol sebacate)-Methacrylate–An Elastomeric Degradable Polymer. Front. Phys., 08 May 2018. https://doi.org/10.3389/fphy.2018.00041

[17] Singh, D.; Harding, A.J.; Albadawi, E.; Boissonade, F.N.; Haycock, J.W.; Claeyssens, F. Additive manufactured biodegradable poly(glycerol sebacate methacrylate) nerve guidance conduits. Acta Biomaterialia 78 (2018) 48–63 49.

[18] Sha, D.; Wu, Z.; Zhang, J.; Ma, Y.; Yang, Z.; Yuan, Y. Development of modified and multifunctional poly(glycerol sebacate) (PGS)-based biomaterials for biomedical applications. European Polymer Journal 161 (2021) 110830

Point 3: It is not clear from the manuscript how does the number of triangles influence the output. Are surface charges from middle pyramids are actually inducing charges on working electrodes?

Response 3: We fabricated the sensor with the number (2, 3, 4, and 5) of the triangular prisms and the open circuit voltages were measured, as plotted in Figure 1. As the number of triangular prisms increased, the voltages were increased and about 1.0 V was generated at 4 or 5 prisms. It shows that the surface charges from middle pyramids increases the potentials generated between the working electrodes. The increase of the generated potentials may be explained via the alignment of a pair of opposite charges in same direction, as shown in Figure 3. Thus, it is expected that as the number of the triangles increased, the generated voltage was also increased.

 We added the following sentences to clarify the influence of the number of triangles and the potential generation mechanism of the sensor to page 4.

“Figure S4 shows the output voltages of the sensor with the number (2, 3, 4, and 5) of the triangular prisms. As the number of triangular prisms increased, the voltages were in-creased and about 1.0 V was generated at 4 or 5 prisms. The increase of the generated po-tentials may be explained via the alignment of a pair of opposite charges in same direction, as shown in Figure S5.”

Figure 1. Open circuit voltages of the sensor with the number (2, 3, 4, and 5) of the triangular prisms

Figure 2. Potential generation mechanism of the sensor

Point 4: It is not clear how parasitic signal from frictional triboelectric charge forming between rubber glove and sensor device was excluded from measurement presented in Fig. 4? Seems that the measured signal from firction between device and rubber is mistaken with device output as it was described here recently: Adv. Mater. 2020, 2002979.

Response 4: We strongly appreciate reviewer’s valuable and helpful comments. We are sorry for the confusion. As the reviewer commented, when a physical contact occurs between rubber glove and sensor device, charges can be transferred and potentials can be generated between them. However, in this manuscript, to avoid it, the PDMS was attached on the glove by using polymer tape, which prevents any interfacial friction between the rubber and the sensor device, as shown in Figure 3. In the paper that the reviewer commented, conductive carbon adhesive was used between the ITO glass plates and the polymer functions for the same effects. Thus, the signals would be not generated between rubber glove and sensor device.

Figure 3. Photo of the sensor attached on the Glove and the schematic image.

Round 2

Reviewer 2 Report

Thank you very much for you good responses. However, I can not accept the statement: “Although the contact-electrification mechanism is not still clear, according to the electron transfer mechanism occurring during metal-insulator contacts, the Al releases electrons to PGSm.” as the electron transfer is not proven during the study. I suggest to modify the discussion by saying that the charged species (which may be everything including electrons) are transfered between chosen materials intensively, as indicated by strong electrical signals. 

Author Response

April 21, 2022

Katarina Nesovic
Assistant Editor
MDPI Belgrade

Manuscript ID: nanomaterials-1657907

Title: 3D multiple triangular prisms for highly-sensitive non-contact mode triboelectric bending sensors

Authors: Gi Hyeon Han, Sun-Woo Kim, Jin-Kyeom Kim, Seung Hyun Lee, Myeong Hoon Jeong, Hyun-Cheol Song, Kyoung Jin Choi, Jeong Min Baik *

Dear Katarina Nesovic,

We believe that the manuscript qualifies as a paper in Nanomaterials. We think that we fully responded to the concerns and comments of the reviewers. Our reasons are as follows:

Reviewer’s comments:

Reviewer#2: Thank you very much for you good responses. However, I can not accept the statement: “Although the contact-electrification mechanism is not still clear, according to the electron transfer mechanism occurring during metal-insulator contacts, the Al releases electrons to PGSm.” as the electron transfer is not proven during the study. I suggest to modify the discussion by saying that the charged species (which may be everything including electrons) are transfered between chosen materials intensively, as indicated by strong electrical signals.

Response 1: We strongly appreciate reviewer’s valuable and helpful comments. According to the reviewer’s comment, we modified the sentence in page 3 as follows;

“This implies that the charged species (which may include electrons) are transferred be-tween two materials intensively, as indicated by strong electrical signals, and the Al acts as a positively charged layer.”

Also, on page 5, we also changed from ‘electrons’ to ‘negative charges’ in a sentence, as follows;

“By the contact between Al and PGSm, negative charges are transferred from Al to PGSm, forming alternating negative and positive charges in turn.”

Sincerely,

Jeong Min Baik

Jeong Min Baik, Co-corresponding author

School of Advanced Materials Science and Engineering

Sungkyunkwan University (SKKU)

Suwon 16419, Republic of Korea

Phone: (82) 31-290-7409
